# Genome-wide association study identifies susceptibility loci for acute myeloid leukemia

Wei-Yu Lin [1], Sarah E. Fordham[1], Eric Hungate[2], Nicola J. Sunter[1], Claire Elstob [1], Yaobo Xu[1], Catherine Park[1], Anne Quante[3,4], Konstantin Strauch[3,4], Christian Gieger [4], Andrew Skol[2], Thahira Rahman[1], Lara Sucheston-Campbell[5], Junke Wang[5], Theresa Hahn [6], Alyssa I. Clay-Gilmour[7], Gail L. Jones[8], Helen J. Marr[8], Graham H. Jackson[8], Tobias Menne[8], Mathew Collin [8], Adam Ivey[9], Robert K. Hills[10], Alan K. Burnett [11], Nigel H. Russell[12], Jude Fitzgibbon [13], Richard A. Larson [2], Michelle M. Le Beau[2], Wendy Stock[2], Olaf Heidenreich[1], Abrar Alharbi[1], David J. Allsup [14], Richard S. Houlston [15], Jean Norden[16], Anne M. Dickinson[16], Elisabeth Douglas[16], Clare Lendrem [16], Ann K. Daly[16], Louise Palm[17], Kim Piechocki[17], Sally Jeffries[17], Martin Bornhäuser[18,19,20], Christoph Röllig[20], Heidi Altmann[20], Leo Ruhnke[20], Desiree Kunadt[20], Lisa Wagenführ[20], Heather J. Cordell [21], Rebecca Darlay[21], Mette K. Andersen[22], Maria C. Fontana[23,24], Giovanni Martinelli[24], Giovanni Marconi [24], Miguel A. Sanz [25,26], José Cervera[25,26], Inés Gómez-Seguí[25,26], Thomas Cluzeau[27], Chimène Moreilhon[27], Sophie Raynaud[27], Heinz Sill [28], Maria Teresa Voso [29], Francesco Lo-Coco[29], Hervé Dombret[30], Meyling Cheok [31], Claude Preudhomme[31], Rosemary E. Gale[32], David Linch[32], Julia Gaal-Wesinger[33], Andras Masszi[34], Daniel Nowak [35], Wolf-Karsten Hofmann[35], Amanda Gilkes[36], Kimmo Porkka[37], Jelena D. Milosevic Feenstra[38], Robert Kralovics[38], David Grimwade[9], Manja Meggendorfer[39], Torsten Haferlach[39], Szilvia Krizsán [40], Csaba Bödör [40], Friedrich Stölzel [20✉], Kenan Onel [41✉] & James M. Allan [1✉]

Acute myeloid leukemia (AML) is a hematological malignancy with an undefined heritable risk. Here we perform a meta-analysis of three genome-wide association studies, with replication in a fourth study, incorporating a total of 4018 AML cases and 10488 controls. We identify a genome-wide significant risk locus for AML at 11q13.2 (rs4930561; $P = 2.15 \times 10^{-8}$; KMT5B). We also identify a genome-wide significant risk locus for the cytogenetically normal AML sub-group (N = 1287) at 6p21.32 (rs3916765; $P = 1.51 \times 10^{-10}$; HLA). Our results inform on AML etiology and identify putative functional genes operating in histone methylation (KMT5B) and immune function (HLA).

A full list of author affiliations appears at the end of the paper.

Acute myeloid leukemia (AML) is the most common acute leukemia in adults of European ancestry[1] comprising distinct sub-groups characterized by unique somatic genetic alterations, etiologies, and outcomes[2]. Rare germline variants in transcription factors and other genes regulating hematopoietic cell differentiation and proliferation are highly penetrant for AML[3], with some causative for debilitating human syndromes that include AML as a component, while others cause familial AML with high penetrance. Although such variants demonstrate a role for genetics in AML susceptibility, they are rare and do not make a major contribution to population disease burden[4]. A modest increased risk of myeloid malignancy for first-degree relatives of non-familial AML patients further supports a role for genetics in disease susceptibility[5], which for the majority of cases is likely determined by co-inheritance of common low penetrance variants consistent with the multigenic model for complex human diseases[6,7].

Outside of highly penetrant syndromic or familial disease, genetic susceptibility to AML remains largely unexplained. To identify AML risk loci we conducted three genome-wide association studies with participants of European ancestry, with replication in a fourth study. Here we report the identification of risk loci for AML, including pan-AML irrespective of disease sub-type and for cytogenetically normal AML. These data inform on disease etiology and demonstrate the existence of common, low-penetrance susceptibility alleles for AML with heterogeneity in risk across sub-types.

## Results

**Discovery meta-analysis of AML genome-wide association studies (GWAS).** We conducted three independent genome-wide association studies with AML cases and controls of European ancestry (GWAS 1-3). Following implementation of rigorous quality metrics to each GWAS (Methods, Supplementary Fig. 1), genotype data were available on 3041 AML cases and 6760 controls of European ancestry (Supplementary Figs. 2–5, Supplementary Table 1). To improve genomic resolution, imputation using the Haplotype Reference Consortium panel (http://www.haplotype-reference-consortium.org/)[8] was used to derive estimated genotypes for >7 million single nucleotide polymorphisms (SNPs) for each study. We combined the association test statistic for 6694056 autosomal SNPs common to all three GWAS after exclusion of those with an INFO (imputation quality) score of <0.6 and a minor allele frequency (MAF) of <0.01, and conducted a meta-analysis under a fixed-effect model for variants associating with AML. Given the genetic and biological heterogeneity of AML, sub-type specific odds ratios (ORs) were also calculated for the largest AML sub-group, cytogenetically normal AML ($N = 822$), using data for 6532175 SNPs common to all three studies. Quantile-quantile plots of observed versus expected $P$ values (minor allele frequency (MAF) >0.01) for all AML cases and cytogenetically normal AML cases showed minimal inflation of test statistics across all three GWAS after adjustment for nominally significant principal components in each GWAS ($\lambda_{GC} = 1.021$, $1.025$, and $1.055$ for all AML in GWAS1, GWAS2, and GWAS3, respectively; $\lambda_{GC} = 1.006$, $1.011$, and $1.025$ for cytogenetically normal in GWAS1, GWAS2, and GWAS3, respectively) (Supplementary Figs. 6–7), minimizing the possibility of hidden population stratification and cryptic relatedness. Pooling data from three GWAS identified associations for cytogenetically normal AML at 6p21.32 (rs3997854, *HLA-DQA2*) and 4q22.3 (rs75391980) that surpassed genome-wide significance ($P \le 5 \times 10^{-8}$), and additional suggestive associations ($P < 10^{-6}$) at 2q36.1 (rs4674579), 3q (rs2621279), 5q35.1 (rs13183143), 11q13.2 (rs11481), and 20p12.3 (rs6077414) for all AML and at 9p21.1 (rs10969985) and 13q21.31 (rs143280565) for

cytogenetically normal AML (Supplementary Fig. 8, Supplementary Table 2).

**Validation GWAS and meta-analysis.** To replicate the associations at the loci identified in the discovery GWAS meta-analysis we conducted a fourth genome-wide association study of European cases and controls (GWAS 4). Following the application of SNP and sample quality control metrics (Supplementary Figs. 9 and 10), data on >7.6 million SNPs from 977 AML cases and 3728 controls of European ancestry were available for analysis, which included 465 cytogenetically normal AML cases (Supplementary Table 1). Quantile–quantile plots of observed versus expected $P$ values (MAF > 0.01) showed minimal inflation of test statistics ($\lambda_{GC} = 1.012$ for all AML cases and $\lambda_{GC} = 1.001$ for cytogenetically normal AML) (Supplementary Fig. 11). Analysis of data from GWAS 4 validated ($P < 0.05$) two of the associations identified in the discovery GWAS meta-analysis, with consistent direction and magnitude of effect sizes across all four studies, including 11q13.2 (rs4930561, *KMT5B*; $P = 2.15 \times 10^{-8}$) for all AML irrespective of sub-type ($N = 4018$) and 6p21.32 (rs3916765, *HLA-DQA2*; $P = 1.51 \times 10^{-10}$) for cytogenetically normal AML ($N = 1287$). Meta-analysis of SNPs common to all four GWAS ($N = 6661818$ and $N = 6496414$ for all AML and cytogenetically normal AML, respectively) also revealed additional borderline significant susceptibility loci at 1p31.1 (rs10789158, *CACHD1*, $P = 2.25 \times 10^{-7}$) for all AML and at 7q33 (rs17773014, *AKR1B1*, $P = 4.09 \times 10^{-7}$) for cytogenetically normal AML, both with consistent direction and magnitude of effect across all four studies (Figs. 1 and 2). In order to test for any potential effects of residual population substructure we re-examined the top signals using only cases and controls of UK origin in GWAS1, GWAS2, and GWAS4, and using only cases and controls of German origin in GWAS3 (Supplementary Fig. 12). The direction and magnitude of the associations in these sub-group analyses (Supplementary Fig. 13) are very similar to the analyses including all cases and controls (Fig. 2). As such, we report genome-wide significant susceptibility loci for all AML and cytogenetically normal AML at 11q13.2 (rs4930561) and 6p21.32 (rs3916765), respectively, and borderline significant susceptibility loci for all AML and cytogenetically normal AML at 1p31.1 (rs10789158) and 7q33 (rs17773014).

AML risk variants at 11q13.2, 6p21.32, 1p31, and 7q33 have Bayesian false-discovery probabilities (BFDP)[9] of 7, <1, 3, and 5%, respectively. There was no evidence of significant heterogeneity ($P < 0.05$) for association with AML for any of the risk variants across the 4 GWAS studies (Fig. 2). Analysis conditioning on the top variant at each susceptibility locus did not identify any evidence of additional associations ($P < 10^{-4}$) within 500 kb of the lead variant (Supplementary Figs. 14–17).

There was no significant heterogeneity in AML risk for any of the four variants when cases and controls were stratified by age (<55 years; ≥55 years) (Supplementary Table 3). Likewise, there was no significant heterogeneity in AML risk for rs4930561 (11q13.2), rs10789158 (1p31.3), or rs17773014 (7q33) when cases and controls were stratified by sex, although there was significant heterogeneity for rs3916765 (6p21.32) with greater penetrance for cytogenetically normal AML in females (OR 2.11, 95% CI 1.63–2.73) compared to males (OR 1.45, 95% CI 1.16–1.80; $I^2 = 78.47$, PQ = 0.03) (Supplementary Table 4).

The relationship between AML risk variants and survival was evaluated in 767 AML patients (excluding acute promyelocytic leukemia) from the UK, Germany, and Hungary. However, none of the 4 AML susceptibility variants identified here were significantly associated with either relapse-free or overall survival in univariate analysis that included all AML patients ($N = 767$) or those with cytogenetically normal AML ($N = 369$) (Supplementary Figs. 18–21).

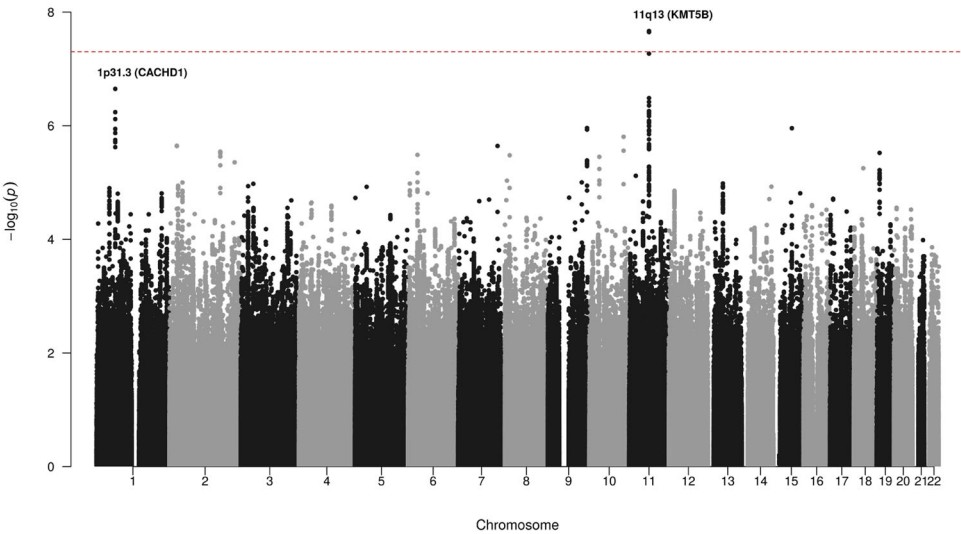

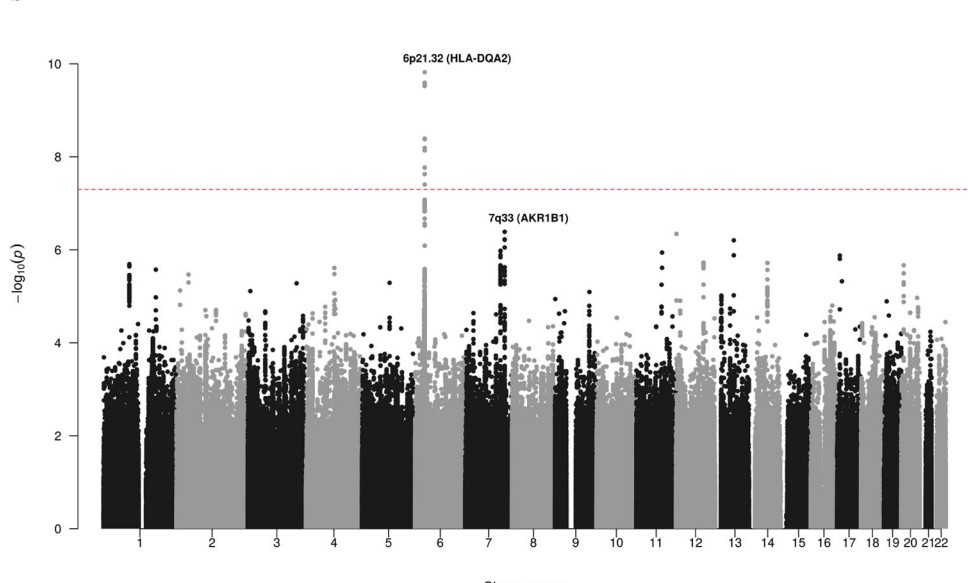

**Fig. 1 Manhattan plots from acute myeloid leukemia meta-analysis of 4 genome-wide association studies.** For each GWAS, association tests were performed for all AML cases and cytogenetically normal AML assuming an additive genetic model, with nominally significant principal components included in the analysis as covariates. Association summary statistics were combined for variants common to all four GWAS, in fixed effects models using PLINK. Manhattan plots show negative $\log_{10}$ (fixed effects meta $P$ values, $Y$ axis) for all AML (**a**) and cytogenetically normal AML (**b**) over 22 autosomal chromosomes. Risk loci are annotated with chromosome position and local gene. All statistical tests were two-sided and no adjustments were made for multiple comparisons. The horizontal red line denotes the threshold for statistical significance in a genome-wide association study ($P < 5.0 \times 10^{-8}$).

A previously reported susceptibility variant for AML in the *BICRA* gene (rs75797233)[10] was not significantly associated with AML risk in our study (GWAS meta-analysis odds ratio (OR) 1.06, 95% CI 0.87–1.29). However, this variant was imputed to sufficient quality (INFO > 0.6) in only three GWAS studies and statistical power to detect an association with AML for this relatively uncommon variant (MAF 0.02) is compromised.

**Inference of risk loci and biological function.** We identified a genome-wide significant association for rs4930561 with risk of AML irrespective of sub-type (OR 1.17, 95% CI 1.11–1.24; $P = 2.15 \times 10^{-8}$)

which maps to the *KMT5B* gene on 11q13.2 (Fig. 3a). To identify putative risk loci we interrogated data from a meta-analysis of 31624 blood samples collated by the eQTLGen consortium[11] for evidence of *cis*-regulated genes. Forty seven genes annotated to within 500 Kb of the association signal and the sentinel variant is a significant eQTL for 12 of these, including *MRPL21* (Benjamini–Hochberg corrected $P$-value $[P_{BH}] = 7.5 \times 10^{-29}$), *RP5-901A4.1* ($P_{BH} = 4.1 \times 10^{-24}$), *ALDH3B1* ($P_{BH} = 4.48 \times 10^{-18}$), *IGHMBP2* ($P_{BH} = 2.85 \times 10^{-16}$), *RP11-802E16.3* ($P_{BH} = 6.76 \times 10^{-15}$), and *CHKA* ($P_{BH} = 1.73 \times 10^{-14}$), *DOC2GP* ($P_{BH} = 1.32 \times 10^{-12}$), *TCIRG1* ($P_{BH} = 7.83 \times 10^{-5}$), *NDUFS8* ($P_{BH} = 1.69 \times 10^{-4}$), *UNC93B1* ($P_{BH} = 1.09 \times 10^{-3}$), *PP1CA* ($P_{BH} = 0.01$), and *RP11-554A11.9* ($P_{BH} = 0.01$). rs4930561 is

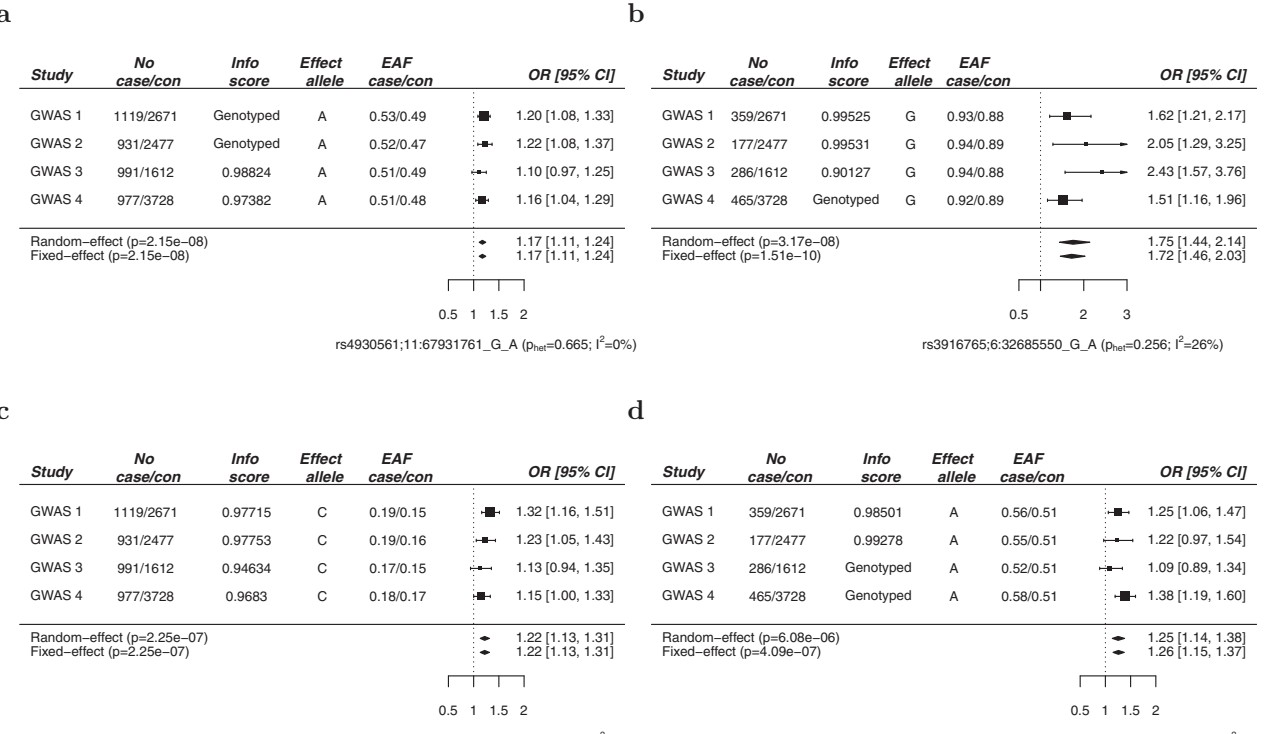

**Fig. 2 Forest plots for 4 new loci associated with acute myeloid leukemia.** Study cohorts, sample sizes (case and controls (con)), imputation (info) score, effect allele, effect allele frequencies (EAF), and estimated odds ratios (OR) for rs4930561 (**a**), rs3916765 (**b**), rs10789158 (**c**), and rs17773014 (**d**). The vertical line corresponds to the null hypothesis (OR = 1). The horizontal lines and square brackets indicate 95% confidence intervals (95% CI). Areas of the boxes are proportional to the weight of the study. Diamonds represent combined estimates for fixed-effect and random-effect analysis. Cochran's Q statistic was used to test for heterogeneity such that $P_{HET} > 0.05$ indicates the presence of non-significant heterogeneity. The heterogeneity index, $I^2$ (0-100) was also measured which quantifies the proportion of the total variation due to heterogeneity. All statistical tests were two-sided and no adjustments were made for multiple comparisons.

not, however, an eQTL for *KMT5B* ($P_{BH} = 0.99$) (Supplementary Table 5).

We identified strong statistical evidence for an association with cytogenetically normal AML for rs3916765 (OR 1.72, 95% CI 1.46–2.03; $P = 1.51 \times 10^{-10}$), which maps to the *HLA* locus on 6p21.32, intergenic to *HLA-DQB1* and *HLA-DQA2* within the major histocompatibility (MHC) class II region (Fig. 3b). To further explore the association between the *HLA* locus and risk of cytogenetically normal AML we imputed classical *HLA* alleles using 5225 Europeans from the Type I Diabetes Genetics Consortium[12] as a reference panel (Methods). *HLA-DQB1*03:02* ($r^2 = 0.56$ with rs3916765) was significantly under-represented in AML cases compared to controls in meta-analysis (OR 0.63, 95% CI 0.51–0.72, $P = 8.9 \times 10^{-8}$), identifying this as the most likely classical *HLA* allele underpinning the association with AML at the 6p21.32 locus (Supplementary Fig. 22). Likewise, *HLA-DQA1*03:01* ($r^2 = 0.53$ with rs3916765) was also significantly under-represented in cases compared to controls (OR 0.77, 95% CI 0.68–0.87, $P = 4.91 \times 10^{-5}$) (Supplementary Fig. 22). However, analysis conditioning on *HLA-DQB1*03:02* rendered the association for the *HLA-DQA1*03:01* allele non-significant (OR 0.98, 95% CI 0.82–1.16, $P = 0.79$), consistent with linkage disequilibrium between these two alleles ($r^2 = 0.50$).

Given the identification of a major AML risk allele at the *HLA* locus on chromosome 6p21.32 and that cancer cells acquire somatic mutations that can function as neo-antigens for immune recognition we performed a case–control analysis stratified by mutation status for *NPM1* and *FLT3*, two genes commonly mutated and clinically significant in cytogenetically normal

AML[2]. Specifically, data on *NPM1* and *FLT3* somatic mutation status were available for 653 and 865 AML cases, respectively. There was no significant heterogeneity in AML risk when cases and controls were stratified by either *NPM1* or *FLT3* mutation status, although there was a trend towards higher risk for *NPM1*-mutated AML (OR 1.96, 95% CI 1.29–2.98; $P = 1.7 \times 10^{-3}$) and *FLT3*-mutated AML (OR 1.52, 95% CI 1.07–2.16; $P = 0.02$) compared to *NPM1*-wildtype AML (OR 1.28, 95% CI 0.97 –1.68; $P = 0.08$) or *FLT3*-wildtpe AML (OR 1.26, 95% CI 1.01–1.58; $P = 0.04$) (Supplementary Table 6).

We also identified a borderline significant association for rs10789158 with AML irrespective of sub-type (OR 1.22, 95% CI 1.13–1.31; $P = 2.25 \times 10^{-7}$) which maps to a block of linkage disequilibrium upstream of the *CACHD1* gene on chromosome 1p31.3 (Fig. 3c). Of the 11 genes annotated to within 500 Kb of the association signal, the sentinel SNP (rs10789158) is eQTL for *RAVER2* ($P_{BH} = 1.26 \times 10^{-2}$) and *AK4* ($P_{BH} = 1.26 \times 10^{-2}$) where the AML risk variant is associated with higher expression of AK4 and lower expression of RAVER2 (Supplementary Table 7).

We also identified a borderline statistically significant association with cytogenetically normal AML for rs17773014 (OR 1.26, 95% CI 1.15–1.37; $P = 4.09 \times 10^{-7}$), which maps close to the *AKR1B1* gene on chromosome 7q33 (Fig. 3d). Nine genes were annotated to within 500 Kb of the association signal and the sentinel SNP (rs17773014) is a significant eQTL for *AKR1B1* in whole blood with the AML risk allele associated with higher transcript levels ($P_{BH} = 5.32 \times 10^{-23}$) (Supplementary Table 8). We cannot exclude an association with related superfamily member *AKR1B10*, which maps close to *AKR1B1* on

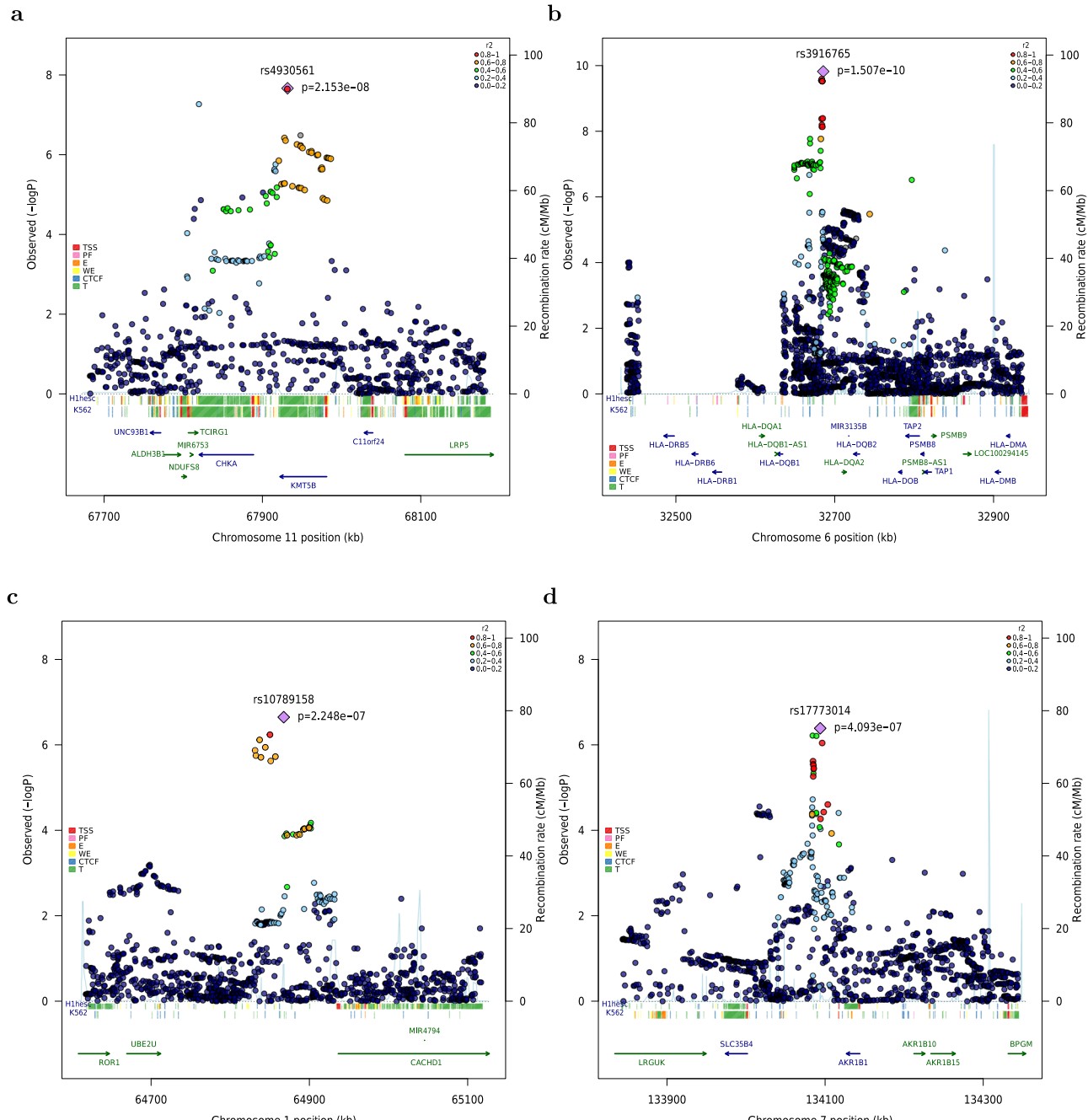

**Fig. 3 Regional association and linkage disequilibrium plots for 4 new loci associated with acute myeloid leukemia.** For each GWAS, association tests were performed for all AML cases and cytogenetically normal AML assuming an additive genetic model, with nominally significant principal components included in the analysis as covariates. Association summary statistics were combined for variants common to all four GWAS, in fixed effects models using PLINK. Negative log$_{10}$-transformed $P$ values (left $Y$ axis) from the meta-analysis of four GWAS are shown for SNPs at 11q13 (**a**) and 1p31.1 (**c**) for all AML irrespective of sub-type, and at 6p21.32 (**b**) and 7q33 (**d**) for cytogenetically normal AML. All statistical tests were two-sided and no adjustments were made for multiple comparisons. The lead SNP at each location is indicated by a purple diamond and the blue line shows recombination rate (right $Y$ axis). All plotted SNPs were either directly genotyped or had an imputation score of >0.6 in all four GWAS datasets. Included are functional annotation tracks from ENCODE showing regulatory activity in H1 human embryonic stem cells (H1hesc) and K562 myeloid leukemia cells. TSS transcription start site, PF promoter flank, E enhancer, WE weak enhancer, CTCF CCCTC-binding factor site, T transcribed.

chromosome 7q33 but which was not annotated in the eQTLGen consortium dataset.

## Discussion
By conducting a meta-analysis of three large genome-wide studies with validation in a fourth study, we identify four susceptibility loci for AML, demonstrating the existence of common, low-penetrance susceptibility alleles for this genetically complex disease. Specifically, our data identify a major susceptibility locus for AML at the 11q13.2 *KMT5B* gene. *KMT5B* (*SUV420H1*) encodes a lysine methyltransferase that is frequently mutated in human cancers, with gene amplifications being particularly common[13–15].

*KMT5B* is implicated in AML pathogenesis where mutation has been associated with transformation from precursor myelodysplastic syndrome to AML[16]. Mutations in other lysine methyltransferases such as *KMT2A* (*MLL1*) occur with high frequency in AML[17]. Although *KMT5B* is a strong candidate for an AML susceptibility gene a priori we cannot exclude mechanisms involving other local genes. For example, the AML risk variant at the 11q13.2 susceptibility locus is significantly associated with lower expression of *CHKA*, which encodes a protein involved in phosphatidylcholine biosynthesis. *CHKA* is significantly upregulated in mouse haematopoietic stem cells and human leukemia cell lines upon restoration of TET2 function[18], a tumor suppressor which blocks aberrant self-renewal and which is frequently mutated in AML resulting in loss of function[19].

We also report a putative pan-AML susceptibility locus at 1p31.3 that is *cis*-eQTL for *RAVER2* and *AK4*. *RAVER2* encodes a ribonucleoprotein involved in RNA splicing where expression associates with transformation from myelodysplastic syndrome to AML[20]. In both human and mouse hematopoietic stem cells *RAVER2* is identified as a target gene for miR-99[21], which regulates normal and malignant hematopoietic stem cell self-renewal[21,22] and expression is significantly associated with prognosis in AML[22]. *AK4* belongs to the adenylate kinase (AK) family of proteins that catalyze the phosphorylation of nucleotide monophosphate precursors to their di- and triphosphate forms. *AK4* is localized to the mitochondria and is a target gene for hypoxia-inducible factor 1 alpha (HIF-1α)[23], an established tumor suppressor in human and murine AML[24,25].

We also identify a major susceptibility locus for cytogenetically normal AML at the 6q21.32 *HLA* gene. This region carries susceptibility alleles for numerous human cancers including hematological malignancies such as chronic lymphocytic leukemia[26,27] and Hodgkin lymphoma[28], where risk is mediated via differential antigenic presentation/T-cell receptor recognition or altered risk of oncogenic infection. Somatic dysregulation of antigenic presentation through altered gene expression, loss of heterozygosity or genomic deletion is a common feature of many human solid cancers leading to failed immune surveillance[29–31]. Somatic loss of HLA alleles is rare in AML at disease presentation although it has been reported as a mechanism of immune escape after bone marrow transplant leading to relapse[32–35]. Our data identify the *HLA-DQB1\*03:02* and *HLA-DQA1\*03:01* alleles as significantly under-represented in cytogenetically normal AML cases compared to controls. Cytogenetically normal AML is characterized by somatic mutation in genes such as *NPM1* and *FLT3*[2] which can function as neo-antigens[36–40]. For example, *NPM1* mutation is reported in up to 60% of cytogenetically normal AML[41,42] where mutation leads to aberrant cytoplasmic expression that is postulated to lead to more efficient HLA presentation[43]. Our data suggest that rs3916765 affects AML risk by modulating immune recognition of mutated cells, although further work is required to determine which leukemia-specific neo-antigens are involved. The *DQB1\*03:02-DQA1\*03:01* haplotype is associated with increased risk of autoimmune diseases including celiac disease[44] and type 1 diabetes[45,46]. Reports of concomitant celiac disease and AML are very rare[47] consistent with the *DQB1\*03:02-DQA1\*03:01* haplotype having pleiotropic and opposing effects on risk of these two diseases. Taken together, these data implicate dysregulated immune function as a risk modifier for cytogenetically normal AML.

Our data suggest that differential expression of *cis*-regulated *AKR1B1* (or related superfamily member *AKR1B10*) at the 7q33 locus modulates risk of cytogenetically normal AML. *AKR1B1* and *AKR1B10* encode members of the aldo-ketoreductase superfamily which catalyze the reduction of numerous aldehydes, including the aldehyde form of glucose to generate fructose via the polyol pathway[48]. Altered glucose metabolism is a hallmark of cancer, where malignant cells switch energy production from mitochondrial oxidative phosphorylation to glycolysis[49]. In AML, a switch to glycolytic metabolism is associated with disease progression and poor outcome[50]. Unlike normal monocytes, AML cells can also utilize fructose as an alternative substrate for glycolysis with expression of the GLUT5 fructose transporter as a major regulator of fructose metabolism in leukemic blast cells[51,52], and where high expression is associated with increased proliferation, clonogenicity, migration, and invasion of AML cells[51]. Interrogation of three independent datasets identified a consistent association between elevated *AKR1B1* transcript levels and shorter overall survival in AML[53] and serum fructose levels are prognostic in AML[51], further implicating this pathway in AML disease progression.

In summary, our study identifies common susceptibility alleles at four genomic locations that modify AML risk, with evidence of sub-type specific risk loci reflecting the existence of multiple etiological pathways to disease development. Further work is required to decipher the functional basis of these risk loci although our data builds on existing evidence demonstrating a role for aberrant histone modification and altered fructose metabolism in AML pathogenesis[16,17,51,52]. Furthermore, the identification of a major AML risk variant at the *HLA* locus on chromosome 6 implicates altered immune function as etiologically important in AML. Our data supports existing evidence of genetic and biological heterogeneity in AML[2] and confirm the need for large collaborative studies to improve statistical power and aid the discovery of sub-type specific genetic risk loci.

## Methods

**Study participants**. GWAS1 comprised 1119 AML cases of European ancestry from UK Medical Research Council/National Cancer Research Institute AML clinical trials ($N = 529$)[54,55], the Eurobank transplant study ($N = 70$)[56], the Leukaemia and Lymphoma Research adult acute Leukaemia population-based case control epidemiology study ($N = 222$)[57,58], and the institutional hematology biobanks of Newcastle University ($N = 48$) and University of Chicago ($N = 250$). Cases were genotyped on the Illumina Omni Express/Omni Express Exome BeadChips. For controls, we used publicly available Illumina Hap550K BeadChip data on 2671 individuals from the British 1958 Birth Cohort[59].

GWAS2 comprised 931 AML cases of European ancestry from the Central England Haemato-Oncology and Oncology Research Bank ($N = 120$) and King's College London Medical School ($N = 144$) with cases recruited from local treatment centers and UK clinical trials. AML cases were also obtained from the Study Alliance Leukemia biobank at Dresden University ($N = 208$)[60,61], the CALGB 9710 APL clinical trial ($N = 101$)[62], the Newcastle University Haematology Biobank ($N = 248$), and the Munich Leukemia Laboratory ($N = 53$) which provides a nationwide diagnostic service in Germany. AML cases were also obtained from treatment centers at the Medical University of Graz ($N = 32$), University Hospital Rigshospitalet Copenhagen ($N = 13$), and the Universita Cattolica Sacro Cuore Rome ($N = 12$). All cases were genotyped on the Illumina Omni Express Exome BeadChip. For controls, we used publicly available Illumina Hap1.2M-Duo data on 2477 individuals from the UK Blood Service Control Group, part of the Wellcome Trust Case Control Consortium 2 study[63].

GWAS3 comprised 991 AML cases of European ancestry from treatment centers at the Department of Hematology and Oncology of the Medical Faculty Mannheim, University of Heidelberg, Germany ($N = 82$)[64], the Centre Hospitalier Universitaire Nice, France ($N = 42$)[65], and the University of London UK (GSE20672) ($N = 40$). AML cases were also obtained from the Austrian Academy of Sciences ($N = 223$)[66], the Acute Leukaemia French Association Clinical Trials at the University of Lille, France ($N = 278$)[67,68], the Biobank La Fe at the Hospital Universitari i Politècnic La Fe, Spain ($N = 15$)[69], the German-Austria AML Study Group the German-Austrian AML Study Group (GSE32462, $N = 189$[70]; GSE34542, $N = 33$[71]; GSE46745, $N = 33$; GSE46951, $N = 51$[72]), and the Cooperative Health Research in the region of Augsburg (KORA) study ($N = 5$)[73]. For controls, we used data from 1612 individuals recruited to KORA study[73]. All genotyping data for cases and controls were generated on the Affymetrix SNP6.0 Array.

GWAS4 (replication) comprised 977 AML cases of European ancestry from the Central England Haemato-Oncology and Oncology Research Bank ($N = 515$), treatment centers at the Hematology Division at Semmelweis University, Budapest ($N = 202$)[74] and the UK Biobank ($N = 260$), a large population-based study conducted in the United Kingdom. For controls, we used 3728 individuals from the UK Biobank. Cases and controls were genotyped on the Affymetrix UK BiLEVE Axiom array or the Affymetrix UK Biobank Axiom array with an equal proportion of cases and controls genotyped on each array.

Collection of patient samples and associated clinico-pathological information was undertaken with written informed consent. All studies were conducted in accordance with the Declaration of Helsinki and received local institutional review board or national research ethics approval (Supplementary Table 9). Specifically, this research has been conducted using the UK Biobank Resource (Application #16583, James Allan). MRC/NCRI AML 11 trial, AML 12 trial and the UK Leukaemia Research Fund (LLR) population-based case–control study of adult acute leukemia received multicenter research ethics committee approval[54,55,75]. Research ethics committee approval was given to the Newcastle Haematology Biobank (07/H0906/109 + 5) and the AML genome-wide association study in the UK (06/q1108/92, BH136664 (7078)). AML cases and controls for Samples from the Hungarian AML patients were obtained during the standard diagnostic workup at the Hematology Divisions of the 1st and 3rd Department of Internal Medicine, Semmelweis University, Budapest, following ethical approval from the Local Ethical Committee (REF TUKEB-1552012) and the Hungarian Medical Research Council (REF 45371-2/2016/EKU). Saliva and fibroblast samples from Austrian AML patients were collected at the Division of Hematology, Medical University of Graz, Graz, Austria[76]. The diagnosis of AML was made in accordance with World Health Organisation guidelines.

**Genotyping and genome-wide quality-control procedures.** Genotype calling was performed using Illumina GenomeStudio software or Affymetrix Genotyping Console software v4.2.0.26. Data handling and analysis was performed using R v3.5.1, PLINK v1.9b4.4, and SNPTEST v2.5.2. Rigorous SNP and sample quality control metrics were applied to all four GWAS (Supplementary Fig. 1). Specifically, we excluded SNPs with extreme departure from Hardy–Weinberg equilibrium (HWE; $P < 10^{-3}$ in either cases or controls) and with a low call rate (<95%). We also excluded SNPs that showed significant differences ($P < 10^{-3}$) between genotype batches and with significant differences ($P < 0.05$) in missingness between cases and controls. Individual samples with a call rate of <95% or with extreme heterozygosity rates (±3 standard deviation from the mean) were also excluded from each GWAS. Individuals were removed such that there were no two individuals with estimated relatedness pihat >0.1875, both within and across GWAS. The individual with the higher call rate was retained unless relatedness was identified between a case and a control, where the case was preferentially retained. Ancestry was assessed using principal component analysis and super-populations from the 1000 genomes project as a reference, with individuals of non-European ancestry excluded based on the first two principal components. In order to minimize any impact of population stratification among the European population we excluded outlying cases and controls identified using principal components 1 and 2 for each GWAS (Supplementary Figs. 1, 3, 4, 5, and 10).

**Imputation, genome-wide association testing, and meta-analysis.** Genome-wide imputation for each GWAS was performed using the Michigan Imputation Server (https://imputationserver.sph.umich.edu/index.html#!) and the Haplotype Reference Consortium reference haplotype panel (http://www.haplotype-reference-consortium.org/) following pre-phasing using ShapeIT (v2.r790)[77]. All variants with an imputation info score <0.6 or a minor allele frequency of <0.01 were excluded from subsequent analysis.

For each GWAS, association tests were performed for all cases and cytogenetically normal AML assuming an additive genetic model, with nominally significant principal components included in the analysis as covariates. Association summary statistics were combined for variants common to GWAS 1, GWAS 2, and GWAS 3, and then for variants common to all four GWAS, in fixed effects models using PLINK v1.9b4.4. Cochran's Q statistic was used to test for heterogeneity and the $I^2$ statistic was used to quantify variation due to heterogeneity.

The Bayesian false discovery probability was calculated using a prior probability of association of 0.0001 and a plausible OR of 1.3[9].

Case–control analyses were also performed stratified by sex and age in all 4 GWAS. For age, cases and controls were stratified into those <55 years and ≥55 years. GWAS 1 was not included in the meta-analysis for the ≥55 age group because the controls were recruited to the 1958 Birth Cohort and were all genotyped at the age of 45 years. Case–control analyses were performed stratified by *NPM1* and *FLT3* mutation status (mutation-positive or mutation-negative) in GWAS 2 and GWAS 4. Data on *NPM1* and *FLT3* somatic mutation status was available for 653 and 865 AML cases, respectively, including 411 and 528 cases of cytogenetically normal AML, respectively. PCR mutation analysis was performed as part of routine diagnostics for *NPM1* exon 12 and *FLT3* exons 14–15 (Supplementary Table 10)[78,79].

**Technical validation of AML susceptibility variants.** All four AML risk variants reported here were either directly genotyped or imputed to high quality. Specifically, rs4930561 was directly genotyped in GWAS 1 and GWAS 2 and imputed in GWAS 3 and GWAS 4 (info score 0.974–0.988); rs3916765 was genotyped in GWAS 4 and imputed in GWAS 1, GWAS 2, and GWAS 3 (info score 0.901–0.995); rs10789158 was imputed in all 4 GWAS studies (info score 0.946–0.9775); and rs17773014 was directly genotyped in GWAS 3 and GWAS 4 and imputed in GWAS 1 and GWAS 2 (info score 0.985–0.993). Fidelity of array genotyping and imputed dosages was confirmed using Sanger sequencing in a

subset of AML samples (including samples genotyped on both Illumina and Affymetrix platforms) for each sentinel variant with perfect or very high concordance for all four variants (Supplementary Figs. 23–26).

The majority of AML cases were genotyped using DNA extracted from cell/tissue samples (blood and bone marrow) taken during AML remission. A minority of AML cases were genotyped using DNA extracted from tissue samples that include leukemic AML cells. As such, we employed a stringent HWE cut-off (Supplementary Fig. 1) in order to eliminate SNPs potentially affected by somatic copy number alterations. Furthermore, we also used Nexus Copy Number v10 (BioDiscovery, California) to interrogate B allele frequency and Log R ratio values at loci associated with AML following genotyping of DNA extracted from leukemic AML cells. For rs4930561 (chromosome 11q13.2) we interrogated data from 352 AML cases using samples with high somatic cell content and found one case with a large deletion capturing the *KMT5B* locus. We also identified 12 cases with evidence of trisomy 11 or large gains affecting chromosome 11, consistent with reports of trisomy 11 in approximately 1% of AML cases[80]. For rs10789158 (chromosome 1p31.3) we identified 1 case with evidence of copy number gain. The susceptibility locus at chromosome 1 does not fall within a region reported to be recurrently somatically deleted or amplified in AML. The association signals at 6p21.32 (rs3916765) and 7q33 (rs17773014) were specific to cytogenetically normal AML and evidence of somatic copy number alterations were visible in 0 and 3 cases, respectively (based on Nexus Copy Number analysis of 127 cytogenetically normal AML cases). Specifically, there were three cases with evidence of deletions affecting the chromosome 7 risk locus that were not visible cytogenetically. Furthermore, there was no evidence of copy neutral loss of heterozygosity (>2 Mb) at any of the four AML susceptibility loci reported here. Taken together, these data limit the possibility of differential genotyping in cases and controls due to somatically acquired allelic imbalance.

**HLA imputation, expression quantitative trait loci (eQTL) analysis, and functional annotation.** Imputation of classical *HLA* alleles was performed using the SNP2HLA v1.0.3 tool using 5225 Europeans from the Type I Diabetes Genetics Consortium as a reference panel[12]. To examine the relationship between SNP genotype and gene expression and identify *cis* expression quantitative trait loci (eQTLs) we made use of data from the eQTLGen Consortium (http://www.eqtlgen.org/cis-eqtls.html) for whole blood. Benjamini–Hochberg (BH)-adjusted $P$ values were estimated for each gene annotated to within 1 Mb of the sentinel SNP at each AML association signal. Regions with AML susceptibility variants were annotated for putative functional motifs using data from the ENCODE project[81].

**Relationship between SNP genotype and patient survival.** The relationship between AML risk variants and survival was evaluated in a total of 767 AML patients (excluding acute promyelocytic leukemia) from the UK[54,55], Germany[60,61], and Hungary[74]. Briefly, patients were treated with conventional intensive AML therapy including ara-C, daunorubicin, and best supportive care. A subset of high-risk patients in the German cohort were treated with stem cell transplantation[60]. Overall survival was defined as the time from diagnosis to the date of last follow-up or death from any cause. Data on relapse-free survival was available on 358 AML patients, which was defined as the time from date of first remission to the date of last follow-up in remission or date of AML relapse. Cox regression analysis was used to estimate allele-specific hazard ratios and 95% confidence intervals for each study in analyses that included all AML cases ($N = 767$) and cytogenetically normal AML ($N = 358$).

**Reporting summary.** Further information on research design is available in the Nature Research Reporting Summary linked to this article.

## Data availability

Genome-wide association summary statistics (Lin_AML_metaassoc.txt) are available for download from https://doi.org/10.25405/data.ncl.16558116.v1. AML case and control genotyping data from the UK Biobank can be obtained via application through https://www.ukbiobank.ac.uk/. Genotyping data on 2699 individuals recruited to the 1958 British Birth Cohort (Hap1.2M-Duo Custom array data) and 2501 individuals from the UK Blood Service are available from the Wellcome Trust Case Control Consortium 2 [https://www.wtccc.org.uk/;WTCCC2:EGAD00000000022,%20EGAD00000000024]. Case and control genotyping data from 1615 individuals recruited to the KORA study can be obtained via application at https://www.helmholtz-muenchen.de/en/kora/. Other genotyping data supporting the findings of this study can be found as deposited in NCBI Gene Expression Omnibus under accession numbers GSE20672, GSE32462, GSE34542, GSE46745, and GSE46951. eQTL data is available from the eQTLGen consortium via http://www.eqtlgen.org/cis-eqtls.html. ENCODE data is available from https://www.encodeproject.org/biosamples/ENCBS718AAA/ for H1 human embryonic stem cells (H1hesc) and from https://www.encodeproject.org/biosamples/ENCBS109ENC/ for K562 myeloid leukemia cells. URLs: Michigan Imputation Server, https://imputationserver.sph.umich.edu/index.html#!; Haplotype Reference Consortium, http://www.haplotype-reference-consortium.org/; eQTLGen Consortium, http://www.eqtlgen.org/cis-eqtls.html; 1000 Genomes Project, https://www.internationalgenome.org/; PLINK, https://www.cog-genomics.org/plink2/; SNPTEST2,

https://www.well.ox.ac.uk/~gav/snptest/; Phenoscanner, http://www.phenoscanner.medschl.cam.ac.uk/;[76] UK Biobank, https://www.ukbiobank.ac.uk/; Central England Haemato-Oncology and Oncology Research Bank, https://bwc.nhs.uk/central-england-haemato-oncology-and-oncology-research-bioba/; Newcastle University Biobank, https://www.ncl.ac.uk/biobanks/; WTCCC, https://www.wtccc.org.uk/; ENCODE, https://www.encodeproject.org/; Study Alliance Leukemia, https://www.sal-aml.org/ueber-uns/einfuehrung-ueberblick.

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

## Acknowledgements

This work was funded by Blood Cancer UK (to JMA; #06002 and #13044). The Hungarian AML study was funded by the Hungarian National Research, Development and Innovation Office (NKFIH) (NVKP_16-1-2016-0004), a Momentum grant (LP- 95021) from the Hungarian Academy of Sciences and EU's Horizon 2020 research and innovation program under grant agreement No. 739593. The KORA study was initiated and financed by the Helmholtz Zentrum München—German Research Center for Environmental Health, which is funded by the German Federal Ministry of Education and Research (BMBF) and by the State of Bavaria. KORA research was supported within the Munich Center of Health Sciences (MC-Health), Ludwig-Maximilians-Universität, as part of LMUinnovativ. The recruitment of cases via the Catholic University of Rome was funded by Progetto AIRC 5permille Mynerva. DN is an endowed Professor of the Deutsche Jose Carreras Leukaemie Stiftung (DJCLS H 03/01) and is funded by the H.W. & J. Hector fund, Baden Wuerttemberg and the Dr. Rolf M. Schwiete Fund, Mannheim. MM is employed by the Munich Leukemia Laboratory and TH is part owner of the Munich Leukemia Laboratory. We are grateful to the Newcastle University Biobank (https://www.ncl.ac.uk/biobanks/) for providing samples.

## Author contributions

W.-Y.L. collated data, conducted data analysis, and drafted the manuscript. S.E.F., N.S., C.E., C.Pa., A.Q., K.S., C.G., L.S.-C., J.W., T.Hah., A.I.C.-G., G.L.J., H.J.M., G.H.J., T.M., M.Co., A.I., R.K.H., A.K.B., N.H.R., J.F., R.A.L., M.M.L.B., W.S., O.H., A.A., D.J.A., R.S.H., J.N., A.M.D., E.D., C.L., A.K.D., L.P., K.Pi., S.J., M.B., C.R., H.A., L.R., D.K., L.W., H.J.C., R.D., M.K.A., M.C.F., G.Mart., G.Marc., M.A.S., J.C., I.G.-S., T.C., C.M., S.R., H.S., M.T.V., F.L.-C., H.D., M.Ch., C.Pr., R.E.G., D.L., J.G.-W., A.M., D.N., W.-K.H., A.G., K.Po., J.D.M.F., R.K., D.G., M.M., T.Haf., S.K., and C.B. collated data and/or advised on data analysis. E.H., Y.X., T.R., and A.S. contributed to preliminary data collation and analysis. F.S. conceived of the project and collated data. K.O. conceived of the project, collated data, and supervised preliminary data analysis. J.M.A. conceived of the project, collated data, analyzed data, directed the research, obtained funding, and drafted the manuscript. All authors approved the final version of the manuscript, with the exception of E.D. (deceased), F.L.-C. (deceased) and D.G. (deceased).

## Funding

## Competing interests

The authors declare no competing interests.

## Additional information

[1]Translational and Clinical Research Institute, Newcastle University Centre for Cancer, Faculty of Medical Sciences, Newcastle University, Newcastle upon Tyne, UK. [2]Section of Pediatric Hematology and Oncology, University of Chicago, Chicago, IL, USA. [3]Helmholtz Zentrum München, German Research Center for Environmental Health, Neuherberg, Germany. [4]Ludwig-Maximilians-Universität München, Chair of Genetic Epidemiology, IBE, Faculty of Medicine, Munich, Germany. [5]College of Pharmacy, The Ohio State University, Columbus, OH, USA. [6]Department of Medicine, Roswell Park Cancer Institute, Buffalo, NY, USA. [7]Arnold School of Public Health, Department of Epidemiology & Biostatistics, University of South Carolina, Greenville, USA. [8]Department of Haematology, Freeman Hospital, Newcastle upon Tyne Hospitals National Health Service Foundation Trust, Newcastle upon Tyne, UK. [9]Department of Medical and Molecular Genetics, King's College Medical School, London, UK. [10]Nuffield Department of Population Health, University of Oxford, Oxford, UK. [11]Paul O'Gorman Leukaemia Research Centre, University of Glasgow, Glasgow, UK. [12]Department of Haematology, Nottingham University Hospitals NHS Trust, Nottingham, UK. [13]Barts Cancer Institute, Queen Mary University of London, London, UK. [14]Centre for Atherothrombosis and Metabolic Disease, Hull York Medical School, Hull, UK. [15]Division of Genetics and Epidemiology, The Institute of Cancer Research, London, UK. [16]Translational and Clinical Research Institute, Faculty of Medical Sciences, Newcastle University, Newcastle upon Tyne, UK. [17]West Midlands Regional Genetics Laboratory, Birmingham Women's Hospital, Birmingham, UK. [18]Department of Haematological Medicine, The Rayne Institute, King's College London, London, UK. [19]National Center for Tumor Diseases NCT, Partner site Dresden, Dresden, Germany. [20]Medizinische Klinik und Poliklinik I, University Hospital Carl Gustav Carus Dresden, Technical University of Dresden, Dresden, Germany. [21]Population Health Sciences Institute, Newcastle University, Newcastle upon Tyne, UK. [22]Department of Clinical Genetics, University Hospital Rigshospitalet, Copenhagen, Denmark. [23]Institute of Hematology "L. and A. Seràgnoli", University of Bologna, Bologna, Italy. [24]IRCCS Istituto Romagnolo per lo Studio dei Tumori (IRST) "Dino Amadori", Meldola, Italy. [25]Hematology Service, Hospital Universitario y Politécnico La Fe, Valencia, Spain. [26]CIBERONC, Instituto de Salud Carlos III, Madrid, Spain. [27]Hematology department, Cote d'Azur University, CHU of Nice, Nice, France. [28]Division of Hematology, Medical University of Graz, Graz, Austria. [29]Università di Roma Tor Vergata, Dipartimento di Biomedicina e Prevenzione, Rome, Italy. [30]Hôpital Saint-Louis, Institut Universitaire d'Hématologie, Université Paris Diderot, Paris, France. [31]Univ. Lille, Inserm, CHU Lille, UMR-S 1172 - JPArc - Centre de Recherche Jean-Pierre AUBERT Neurosciences et Cancer, F-59000 Lille, France. [32]Department of Haematology, University College London Cancer Institute, London, UK. [33]1st Department of Internal Medicine, Semmewleis University, Budapest, Hungary. [34]3rd Department of Internal Medicine, Semmewleis University, Budapest, Hungary. [35]Department of Hematology and Oncology, Medical Faculty Mannheim, Heidelberg University, Mannheim, Germany. [36]Department of Haematology, University of Cardiff, Cardiff, UK. [37]Helsinki University Hospital Comprehensive Cancer Center, Hematology Research Unit Helsinki, University of Helsinki, Helsinki, Finland. [38]Department of Laboratory Medicine, Medical University of Vienna, Vienna, Austria. [39]MLL Munich Leukemia Laboratory, Munich, Germany. [40]HCEMM-SE Molecular Oncohematology Research Group, 1st Department of Pathology and Experimental Cancer Research, Semmelweis University, Budapest, Hungary. [41]Department of Genetics and Genomic Sciences, Icahn School of Medicine at Mount Sinai, New York, NY, USA. ✉email: friedrich.stoelzel@uniklinikum-dresden.de; kenan.onel@mssm.edu; james.allan@newcastle.ac.uk

