## [Peer Review File · Nature Communications]

Genome-wide association study identifies susceptibility loci for acute myeloid leukemiaReviewers' comments:

Reviewer #1 (Remarks to the Author); expert on GWAS:

This study is a genome-wide association study for acute myeloid leukemia (AML), which is a relatively rare cancer with little known about its etiology. The authors meta-analyzed 4 GWAS for AML and found 3 genome-wide significant variants for cytogenetically normal AML and 1 suggestive finding for AML overall. Although they had ~4400 cases and 10,500 controls for AML overall, the genome-wide significant findings observed were for cytogenetically normal AML cases, where they had fewer cases (n=920 for GWAS1-3 and n=472? from GWAS4). (It would be helpful if the total number of cytogenetically normal cases were provided in the abstract/manuscript.) The study is interesting and represents a large undertaking, but I have major concerns about some of the methods, particularly the controls used for this study, which do not seem well matched to cases. I am concerned that some of the findings could be false-positives due to population stratification. More specific comments are below:

1. One of the major concerns that I have with this manuscript is the use of convenience controls, which I am not convinced are genetically well matched to the cases. For GWAS 1, GWAS2, and GWAS4, the authors used previously genotyped controls from the U.K.; however, some of their cases in all of these GWAS came from other countries, such as the Budapest, U.S., Germany, Italy, and Denmark. GWAS3 is also problematic, because it has German controls, but cases from France, Italy and Spain. I don't think that British or German controls are appropriate for many of these cases from other countries. The authors should either limit their analyses to cases and controls from the same country or find appropriate controls for cases in different countries.
2. The authors provide supplementary plots showing the principal components for GWAS1, GWAS2, and GWAS3 (Sup Figures 2-4) and GWAS 4 (Sup Fig 13), but these are insufficient. The plots are difficult to evaluate because they include samples from other ancestries from 1000 genomes. All four plots look similar, because they all contain the same 1KG samples from 3 ancestries and it is hard to see the distribution of the cases and controls, which is most important. To me, the cases and controls do not appear very well matched for GWAS1-3. There is some overlap between the cases and controls, but the cases are clearly shifted to the right in all three plots. GWAS4 looks better, but it is still hard to evaluate. The authors should construct the PCs using only their cases and controls for each GWAS separately and provide plots of the PCs for each of the 4 GWAS. Ideally, they should provide plots of the first 5 or 7 principal components, not just PC1 vs PC2, for each GWAS. They should consider genetically matching cases and controls. Finally, for each GWAS, were the first 10 principal components tested in a null model without genotypes for AML overall and cytogenetically normal AML (CN-AML)? What were those results and were all principal components that were nominally significantly associated with case-control status in the null model included for adjustment in the association testing?
3. The authors provide some technical validation, but presumably this was only done on the cases (not controls, which were often genotyped on another array) and no details of the number of samples were provided or what geographical region/center they were from.
4. It would be helpful if the authors could provide a supplementary table with the number of cytogenetically normal AML cases and other AML cases from each center for each GWAS and some details of the case ascertainment (e.g., clinical trial, etc...).
5. The authors imputed the classical HLA alleles for one locus. What was the correlation between the two HLA alleles with the strongest association and was all of the risk explained by HLA-DQB*03:02 in the conditional analysis?
6. It is not clear if other prognostic risk factors were considered in the survival analysis. Some additional details about the cases used for survival analysis would be helpful. Were they all CN-AML cases? The subset of cases treated with stem cell transplantation seems uniquely different; its not clear why they were included.
7. Were molecular subgroups of CN-AML explored? Supplementary Table 7 is hard to interpret. The authors state that it is a case-case analysis, but it is unclear what age threshold they used to stratify their cases. Supplementary Table 8 is also hard to interpret. It would be clearer if the authors provided ORs stratified by sex and provided the number of male and female cases/controls.

Reviewer #2 (Remarks to the Author); expert on AML genomics:

In this study, Lin and colleagues have performed four genome-wide association studies incorporating a total of 4413 AML cases and 10542 controls. They identified three risk loci for cytogenetically normal AML at 6p21.32, 7q33, and 17p13.1, as well as an association for AML irrespective of subtype at 1p31.3, which are informative on disease etiology.

Major comments:

- With regard to the pooled GWAS data, did the authors look also for subgroups in addition to cytogenetically normal AML, i.e. for example AML with complex genomic aberrations or AML with core binding factor abnormalities?
- Similarly, it would be important to have additional molecular genomic information, at least on the AML subclass defining aberrations NPM1 and CEBPA. These markers define distinct WHO / ELN AML subgroups and it would be of great interest to know whether the risk loci identified for cytogenetically normal AML were associated with one of these subgroups.
- In the last part of the results section, the authors state that they did not find any significant association with patient survival. Please comment, why data on outcome/treatment was only available in 767 out of 4413 AML patients?
- Given the heterogeneous treatment of patients, I would have assumed that it will be difficult to identify loci associated with overall survival. However, to the reader it would be of great interest whether loci could be identified that are associated with response to cytarabine or daunorubicin, the standard chemotherapy backbone used in almost all patients who can be intensively treated. Previous reports in childhood AML suggested the presence of respective loci (Bargal et al. Oncotarget 2018).

Minor comments:

- In the abstract, the authors state that their findings "identify AML as a genetically heterogeneous malignancy". In my opinion, this was already well known prior to this study. In accordance, I would suggest to rephrase this sentence to "confirm AML as a genetically heterogeneous malignancy".

Lin et al. Genome-wide association study identifies four susceptibility loci for acute myeloid leukemia

Thank you for giving us the opportunity to respond to reviewers' comments and revise our manuscript accordingly. The reviewers have made excellent suggestions that have significantly strengthened our manuscript. We have addressed each of the reviewer's comments as detailed below and changes to the text are highlighted in red.

Reviewer #1 (Remarks to the Author); expert on GWAS:

Introductory comment: This study is a genome-wide association study for acute myeloid leukemia (AML), which is a relatively rare cancer with little known about its etiology. The authors meta-analyzed 4 GWAS for AML and found 3 genome-wide significant variants for cytogenetically normal AML and 1 suggestive finding for AML overall. Although they had ~4400 cases and 10,500 controls for AML overall, the genome-wide significant findings observed were for cytogenetically normal AML cases, where they had fewer cases (n=920 for GWAS1-3 and n=472? from GWAS4). (It would be helpful if the total number of cytogenetically normal cases were provided in the abstract/manuscript.) The study is interesting and represents a large undertaking, but I have major concerns about some of the methods, particularly the controls used for this study, which do not seem well matched to cases. I am concerned that some of the findings could be false-positives due to population stratification. More specific comments are below:

Author response to introductory comment: We have now included the number of cytogenetically normal cases to the Abstract and throughout the manuscript, as follows:

Abstract:

Acute myeloid leukemia (AML) is a hematological malignancy with an undefined heritable risk. Here we performed a meta-analysis of three genome-wide association studies, with replication in a fourth study, incorporating a total of 3918 AML cases and 10488 controls. We identified a genome-wide significant risk locus for AML at 11q13.2 (rs4930561; $P = 2.15 \times 10^{-8}$; *KMT5B*) and a borderline significant association at 1p31.3 (rs10789158; $P = 2.25 \times 10^{-7}$; *CACHD1*). We also identified a genome-wide significant risk locus for the cytogenetically normal AML sub-group (N=1287) at 6p21.32 (rs3916765; $P = 1.51 \times 10^{-10}$; *HLA*) and a borderline significant association at 7q33 (rs17773014; $P = 4.09 \times 10^{-7}$; *AKR1B*). Our results inform on AML etiology by identifying putative functional genes operating in histone methylation (*KMT5B*), immune function (*HLA*), stemness (*CACHD1*) and fructose metabolism (*AKR1B*).

We have also added a new supplementary table showing the demographics of the AML cases in all four GWAS studies (Supplementary Table 1).

The reviewer also raises a very important point regarding population stratification and potential case-control bias, which is also the subject of comments 1 and 2, as follows:

Comment 1. One of the major concerns that I have with this manuscript is the use of convenience controls, which I am not convinced are genetically well matched to the cases. For GWAS 1, GWAS2,

and GWAS4, the authors used previously genotyped controls from the U.K.; however, some of their cases in all of these GWAS came from other countries, such as the Budapest, U.S., Germany, Italy, and Denmark. GWAS3 is also problematic, because it has German controls, but cases from France, Italy and Spain. I don't think that British or German controls are appropriate for many of these cases from other countries. The authors should either limit their analyses to cases and controls from the same country or find appropriate controls for cases in different countries.

Comment 2. The authors provide supplementary plots showing the principal components for GWAS1, GWAS2, and GWAS3 (Sup Figures 2-4) and GWAS 4 (Sup Fig 13), but these are insufficient. The plots are difficult to evaluate because they include samples from other ancestries from 1000 genomes. All four plots look similar, because they all contain the same 1KG samples from 3 ancestries and it is hard to see the distribution of the cases and controls, which is most important. To me, the cases and controls do not appear very well matched for GWAS1-3. There is some overlap between the cases and controls, but the cases are clearly shifted to the right in all three plots. GWAS4 looks better, but it is still hard to evaluate. The authors should construct the PCs using only their cases and controls for each GWAS separately and provide plots of the PCs for each of the 4 GWAS. Ideally, they should provide plots of the first 5 or 7 principal components, not just PC1 vs PC2, for each GWAS. They should consider genetically matching cases and controls. Finally, for each GWAS, were the first 10 principal components tested in a null model without genotypes for AML overall and cytogenetically normal AML (CN-AML)? What were those results and were all principal components that were nominally significantly associated with case-control status in the null model included for adjustment in the association testing?

Author response to comments 1 and 2: The reviewer is absolutely correct in that population stratification should be appropriately adjusted for by inclusion of significant principal components as co-variables in the association analysis.

On re-reading our manuscript it is clear that we did not include sufficient detail on this critical aspect of our study and that we should have included PCA plots showing only cases and controls, as noted by the reviewer. As such, we have now included PCA plots for the first 10 principal components showing only cases and controls for each GWAS (Supplementary Figures 3, 4, 5 and 10). We can also confirm that all nominally significant ($P < 0.05$) principal components were included for adjustment in the association testing. Furthermore, in response to the reviewer comment regarding population stratification we have added an additional step to our data cleaning pipeline. Specifically, on review of the original PCA plots we identified a small number of outlying cases and controls (based on the first two principal components for each GWAS) which we have now removed. The result of this additional quality control step is that there is improved cases/controls matching, as demonstrated by the PCA plots showing that they occupy the same space for the first ten principal components. Furthermore, the genomic inflation factors for all four GWAS are low for analyses including all AML cases ($\lambda_{GC} = 1.012-1.055$) and cytogenetically normal AML ($\lambda_{GC} = 1.001-1.025$), minimising the possibility of hidden population stratification or genotyping errors between cases and controls.

We have now included additional text describing our approach in the Methods and Results sections of the revised manuscript and cited the genomic inflation factor for each analysis, as follows:

Methods

Ancestry was assessed using principal component analysis and super-populations from the 1000 genomes project as a reference, with individuals of non-European ancestry excluded based on the first two principal components. **In order to minimise any impact of population stratification among the European populations, we excluded outlying cases and controls identified using principal components 1 and 2 for each GWAS (Supplementary Figures 1, 3, 4, 5 and 10).**

For each GWAS, association tests were performed for all cases and cytogenetically normal AML assuming an additive genetic model, with **nominally significant principal components included in the analysis as covariates**. Association summary statistics were combined for variants common to GWAS 1, GWAS 2 and GWAS 3, and then for variants common to all four GWAS, in fixed effects models using PLINK. Cochran's Q statistic was used to test for heterogeneity and the I^2 statistic was used to quantify variation due to heterogeneity.

Results

Quantile-quantile plots of observed versus expected P values (minor allele frequency (MAF) > 0.01) for all AML cases and cytogenetically normal AML cases showed minimal inflation of test statistics across all three GWAS **after adjustment for nominally significant principal components in each GWAS ($\lambda_{GC} = 1.021, 1.025$ and 1.055 for all AML in GWAS1, GWAS2 and GWAS3, respectively; $\lambda_{GC} = 1.006, 1.011$ and 1.025 for cytogenetically normal in GWAS1, GWAS2 and GWAS3, respectively)** (Supplementary Figures 6-7), minimising the possibility of hidden population stratification and cryptic relatedness.

To replicate the associations at the loci identified in the discovery GWAS meta-analysis we conducted a fourth genome-wide association study of European cases and controls (GWAS 4). Following the application of SNP and sample quality control metrics (Supplementary Figures 9 and 10), data on > 7.6 million SNPs from **977** AML cases and **3728** controls of European ancestry were available for analysis, which included 465 cytogenetically normal AML cases (Supplementary Table 1). Quantile-quantile plots of observed versus expected P values (MAF > 0.01) showed minimal inflation of test statistics ($\lambda_{GC} = 1.012$ for all AML cases and $\lambda_{GC} = 1.001$ for cytogenetically normal AML) (Supplementary Figure 11).

We hope that these changes will now enable the reader to easily see the structure of the case and control populations in all four GWAS more clearly.

The exclusion of outlying cases and controls has reduced the statistical power of the GWAS meta analysis such that the significance for all 4 hits reported in our original manuscript has slightly reduced. Despite this, three of the results are retained in the revised manuscript with the 6p21 (HLA) hit reaching the GWAS threshold ($P = 1.51 \times 10^{-10}$) and the 1p31 and 7q33 hits reported as borderline significant hit ($P < 5 \times 10^{-7}$). The P value for the fourth hit on chromosome 17 has dropped below the reportable threshold and has been removed from the revised manuscript. Conversely, the additional data cleaning has pushed a signal on chromosome 11 over the GWAS threshold. This hit (for all AML regardless of sub-type) approached the reportable threshold in our original dataset but now reaches the GWAS threshold ($P = 2.15 \times 10^{-8}$) and has been included in the revised manuscript, as follows:

Results

We identified a genome-wide significant association for rs4930561 with risk of AML irrespective of sub-type (OR 1.17, 95% CI 1.11-1.24; $P = 2.15 \times 10^{-8}$) which maps to the *KMT5B* gene on 11q13.2

(Figure 3A). To identify putative risk loci we interrogated data from a meta-analysis of 31624 blood samples collated by the eQTLGen consortium¹¹ for evidence of *cis*-regulated genes. **Fourty seven genes annotated to within 500Kb of the association signal and the sentinel variant is a significant eQTL for 12 of these, including *MRPL21* (Benjamini-Hochberg corrected *P*-value [P_{BH}] = 7.5×10^{-29}), *RP5-901A4.1* ($P_{BH} = 4.1 \times 10^{-24}$), *ALDH3B1* ($P_{BH} = 4.48 \times 10^{-18}$), *IGHMBP2* ($P_{BH} = 2.85 \times 10^{-16}$), *RP11-802E16.3* ($P_{BH} = 6.76 \times 10^{-15}$) and *CHKA* ($P_{BH} = 1.73 \times 10^{-14}$), *DOC2GP* ($P_{BH} = 1.32 \times 10^{-12}$), *TCIRG1* ($P_{BH} = 7.83 \times 10^{-5}$), *NDUFS8* ($P_{BH} = 1.69 \times 10^{-4}$), *UNC93B1* ($P_{BH} = 1.09 \times 10^{-3}$), *PP1CA* ($P_{BH} = 0.01$) and *RP11-554A11.9* ($P_{BH} = 0.01$). rs4930561 is not, however, an eQTL for *KMT5B* ($P_{BH} = 0.99$)(Supplementary Table 5).**

Discussion

By conducting a meta-analysis of three large genome-wide studies with validation in a fourth study, we identify four susceptibility loci for AML, demonstrating the existence of common, low-penetrance susceptibility alleles for this genetically complex disease. **Specifically, our data identify a major susceptibility locus for AML at the 11q13.2 *KMT5B* gene. *KMT5B* (*SUV420H1*) encodes a lysine methyltransferase that is frequently mutated in human cancers, with gene amplifications being particularly common^{13, 14, 15}. *KMT5B* has been implicated in AML pathogenesis where mutation has been associated with transformation from precursor myelodysplastic syndrome to AML¹⁶. Mutations in other lysine methyltransferases such as *KMT2A* (*MLL1*) occur with high frequency in AML¹⁷. Although *KMT5B* is a strong candidate for an AML susceptibility gene *a priori* we cannot exclude mechanisms involving other local genes. For example, the AML risk variant at the 11q13.2 susceptibility locus is significantly associated with lower expression of *CHKA*, which encodes a protein involved in phosphatidylcholine biosynthesis. *CHKA* is significantly upregulated in mouse haematopoietic stem cells and human leukemia cell lines upon restoration of TET2 function¹⁸, a tumour suppressor which blocks aberrant self-renewal and which is frequently mutated in AML resulting in loss of function¹⁹.**

Comment 3. The authors provide some technical validation, but presumably this was only done on the cases (not controls, which were often genotyped on another array) and no details of the number of samples were provided or what geographical region/center they were from.

Author response: Unfortunately, it has not been possible to access material or sequencing data from controls. Specifically, whole-genome sequencing data is not available for the WTCCC controls (Illumina) (personal communication Dr. Nicholas Watkins, Assistant Director, UK Blood Service). The UK Biobank are currently undertaking whole genome sequencing on participants, but these data are not yet available. However, we have included Sanger sequencing validation on additional AML cases and we have also included samples genotyped on the Illumina platform as well as the Affymetrix platform. The results of these validation experiments, including representative examples of each genotype, are shown in Supplementary Figures 21-24 and described in the Methods, as follows:

Materials and Methods

All four AML risk variants reported here were either directly genotyped or imputed to high quality. Specifically, **rs4930561 was directly genotyped in GWAS 1 and GWAS 2 and imputed in GWAS 3 and GWAS 4 (info score 0.974 - 0.988)**; rs3916765 was genotyped in GWAS 4 and imputed in GWAS 1, GWAS 2 and GWAS 3 (info score 0.901-0.995); rs10789158 was imputed in all 4 GWAS studies (info score 0.946-0.9775); and rs17773014 was directly genotyped in GWAS 3 and GWAS 4 and imputed in GWAS 1 and GWAS 2 (info score 0.985-0.993). Fidelity of array genotyping and imputed dosages was confirmed using Sanger sequencing in a subset of AML samples **(including samples genotyped on both**

Illumina and Affymetrix platforms) for each sentinel variant with perfect or very high concordance for all four variants (Supplementary Figures 21 - 24).

4. It would be helpful if the authors could provide a supplementary table with the number of cytogenetically normal AML cases and other AML cases from each center for each GWAS and some details of the case ascertainment (e.g., clinical trial, etc...).

Author response: We have also added a new supplementary table showing the demographics of AML cases in all four GWAS studies (Supplementary Table 1). We have also included information on the case ascertainment from each centre in the Methods along with citations to relevant publications, as follows:

Methods

GWAS1 comprised **1119** AML cases of European ancestry from **UK Medical Research Council/National Cancer Research Institute AML clinical trials (N=528)**^{52, 53}, the Eurobank transplant study (N=70)⁵⁴, the Leukaemia and Lymphoma Research **adult acute Leukaemia population-based case control epidemiology study (N=223)**^{55, 56}, and the **institutional haematology biobanks of Newcastle University (N=48) and University of Chicago (N=250)**. Cases were genotyped on the Illumina Omni Express/Omni Express Exome BeadChips. For controls, we used publicly available Illumina Hap550K BeadChip data on 2671 individuals from the British 1958 Birth Cohort⁵⁷.

GWAS 2 comprised **931** AML cases of European ancestry from the Central England Haemato-Oncology and Oncology Research Bank (N=**120**) and King's College London Medical School (N=144) **with cases recruited from local treatment centres and UK clinical trials. AML cases were also obtained from the Study Alliance Leukemia biobank at Dresden University (N=208)**^{58, 59}, the CALGB 9710 APL **clinical trial (N=101)**⁶⁰, the Newcastle University Haematology Biobank (N=**248**) and the Munich Leukemia Laboratory (N=**53**) **which provides a nationwide diagnostic service in Germany. AML cases were also obtained from treatment centres at the** Medical University of Graz (N=**32**), University Hospital Rigshospitalet Copenhagen (N=13) and the Università Cattolica Sacro Cuore Rome (N=**12**). All cases were genotyped on the Illumina Omni Express Exome BeadChip. For controls, we used publicly available Illumina Hap1.2M-Duo data on **2477** individuals from the UK Blood Service Control Group, part of the Wellcome Trust Case Control Consortium 2 controls⁶¹.

GWAS 3 comprised **991** AML cases of European ancestry **from treatment centres at** the Department of Hematology and Oncology of the Medical Faculty Mannheim, University of Heidelberg, Germany (N=82)⁶², the Centre Hospitalier Universitaire Nice, France (N=**42**)⁶³ and the University of London UK (GSE20672)(N=**40**). AML cases were also obtained from **the ALFA-0701 clinical trial at** the Austrian Academy of Sciences (N=**223**)⁶⁴, the **Acute Leukaemia French Association 9801/9802 Clinical Trials** at the University of Lille, France (N=**278**)^{65, 66}, **the Biobank La Fe** at the Hospital Universitari i Politènic La Fe, Spain (N=**15**)⁶⁷, the German-Austria AML Study Group the German-Austrian AML Study Group (GSE32462, N=**189**⁶⁸; GSE34542, N=**33**⁶⁹; GSE46745, N=33; GSE46951, N=51⁷⁰), and the Cooperative Health Research in the region of Augsburg (KORA) study (N=5)⁷¹. For controls, we used data from **1612** individuals recruited to KORA study⁷¹. All genotyping data for cases and controls were generated on the Affymetrix SNP6.0 Array.

GWAS4 (replication) comprised **977** AML cases of European ancestry from the Central England Haemato-Oncology and Oncology Research Bank (N=**515**), **treatment centres at the** Hematology Division at Semmelweis University, Budapest (N=**202**)⁷² and the UK Biobank (N=**260**), a large population-based study conducted in the United Kingdom. For controls, we used **3728** individuals from the UK Biobank. Cases and controls were genotyped on the Affymetrix UK BiLEVE Axiom array

or the Affymetrix UK Biobank Axiom array with an equal proportion of cases and controls genotyped on each array.

5. The authors imputed the classical HLA alleles for one locus. What was the correlation between the two HLA alleles with the strongest association and was all of the risk explained by HLA-DQB*03:02 in the conditional analysis?

*Author response: As suggested by the reviewer, we have now performed an association analysis conditioning on HLA-DQB*03:02 and reported the results in supplementary figure 20 and the manuscript text, as follows:*

Results

Likewise, HLA-DQA1*03:01 ($r^2 = 0.53$ with rs3916765) was also significantly under-represented in cases compared to controls (OR 0.77, 95% CI 0.68-0.87, $P = 4.91 \times 10^{-5}$) (Supplementary Figure 20). However, analysis conditioning on HLA-DQB1*03:02 rendered the association for the HLA-DQA1*03:01 allele non-significant (OR 0.98, 95% CI 0.82-1.16, $P = 0.79$), consistent with linkage disequilibrium between these two alleles ($r^2 = 0.50$).

6. It is not clear if other prognostic risk factors were considered in the survival analysis. Some additional details about the cases used for survival analysis would be helpful. Were they all CN-AML cases? The subset of cases treated with stem cell transplantation seems uniquely different; its not clear why they were included.

Author response: The post-treatment survival analysis included all patients with available survival data regardless of karyotype in a univariate analysis. Had any of the risk variants been significantly associated with survival in univariate analysis then we would have conducted a multivariate analysis with established prognostic markers to test for independence. In response to the reviewers comment we have now also included analysis for relapse-free and overall survival restricted to those patients with cytogenetically normal AML (Supplementary Figure 19) and updated the text in the Methods and Results as follows:

Methods

Cox regression analysis was used to estimate allele specific hazard ratios and 95% confidence intervals for each study in analyses that included all AML cases (N=767) and cytogenetically normal AML (N=369).

Results

The relationship between AML risk variants and survival was evaluated in 767 AML patients (excluding acute promyelocytic leukemia) from the UK, Germany and Hungary. However, none of the 4 AML susceptibility variants identified here were significantly associated with either relapse-free or overall survival in univariate analysis that included all AML patients or those with cytogenetically normal AML (N=369)(Supplementary Figures 16 – 19).

The reviewer is correct in that one of the cohorts with available survival data includes patients who were transplanted. Given that the HLA locus carries a risk allele for AML and that HLA genotype is a major determinant of outcome after transplant we decided to retain this cohort in the analysis, although the number of patients transplanted is too low to warrant a specific analysis of this sub-group.

7. Were molecular subgroups of CN-AML explored? Supplementary Table 7 is hard to interpret. The authors state that it is a case-case analysis, but it is unclear what age threshold they used to stratify their cases. Supplementary Table 8 is also hard to interpret. It would be clearer if the authors provided ORs stratified by sex and provided the number of male and female cases/controls.

Author response: Regarding molecular subgroups, a similar comment was also made by reviewer 2 (see comment 2). In response, we have collated data on NPM1 and FLT3 mutation status and included analyses with cases and controls stratified by mutation status – please see our response to reviewer 2 comment 2 for additional details.

The data originally presented in Supplementary Table 7 included age as a continuous variable in a case-only analysis, which we agree was rather confusing. As such, we have now included analyses with cases and controls stratified by age group (age <55 and ≥55 years) for all four hits, and report the results in supplementary table 3 and in the text as follows:

Materials and Methods

Case-control analyses were also performed stratified by sex and age in all 4 GWAS. For age, cases and controls were stratified into those < 55 years and ≥ 55 years. GWAS1 was not included in the meta-analysis for the ≥ 55 age group because the controls were recruited to the 1958 Birth Cohort and were all genotyped at the age of 45 years.

Results

There was no significant heterogeneity in AML risk for any of the four variants when cases and controls were stratified by age (< 55 years; ≥ 55 years) (Supplementary Table 3).

In response to the reviewers comments we have also included analyses with cases and controls stratified by sex for all four hits, and report the results in supplementary table 4 and in the text as follows:

Likewise, there was no significant heterogeneity in AML risk for rs4930561 (11q13.2), rs10789158 (1p31.3) or rs17773014 (7q33) when cases and controls were stratified by sex, although there was significant heterogeneity for rs3916765 (6p21.32) with greater penetrance for cytogenetically normal AML in females (OR 2.11, 95% CI 1.63-2.73) compared to males (OR 1.45, 95% CI 1.16-1.80; $I^2 = 78.47$, $PQ = 0.03$)(Supplementary Table 4).

Reviewer #2 (Remarks to the Author); expert on AML genomics:

In this study, Lin and colleagues have performed four genome-wide association studies incorporating a total of 4413 AML cases and 10542 controls. They identified three risk loci for cytogenetically normal AML at 6p21.32, 7q33, and 17p13.1, as well as an association for AML irrespective of subtype at 1p31.3, which are informative on disease etiology.

Major comments:

Comment 1: With regard to the pooled GWAS data, did the authors look also for subgroups in addition to cytogenetically normal AML, i.e. for example AML with complex genomic aberrations or AML with core binding factor abnormalities?

Author response: We have collated data on other major cytogenetic sub-groups and there is suggestive evidence of sub-type specific risk alleles for t(15;17) acute promyelocytic leukemia, core-binding-factor AML and complex karyotype AML. However, none of these signals currently reach the genome-wide threshold for statistical significance required for reporting, primarily due to the limited size of these analyses. As such, we have recently applied for funding to further expand our study and expect to identify sub-type specific risk alleles for these leukemias in the future.

Comment 2: Similarly, it would be important to have additional molecular genomic information, at least on the AML subclass defining aberrations NPM1 and CEBPA. These markers define distinct WHO / ELN AML subgroups and it would be of great interest to know whether the risk loci identified for cytogenetically normal AML were associated with one of these subgroups.

Author response: One hypothesis explaining the observed association between HLA genotype and AML risk is that leukemic cells acquire somatic alterations that function as neo-antigens for immune recognition, which is alluded to by the reviewer. Cytogenetically normal AML can be further sub-grouped by acquired somatic mutations with many of these giving rise to immunogenic HLA-presented peptides, including those derived from mutant NPM1 and mutant FLT3. Given this, we have collated FLT3 and NPM1 mutation data on >500 cases recruited to the study, and have performed sub-group analysis for the HLA risk variant associated with cytogenetically normal AML stratified by NPM1 and FLT3 mutation status. Unfortunately, it has not been possible to collate sufficient data for CEBPA and the analyses are somewhat underpowered for NPM1 and FLT3, but worthy of inclusion nonetheless, as suggested by the reviewer. We have presented these data in supplementary table 6 and included new text in the methods and results section, as follows:

Methods

Case-control analyses were performed stratified by *NPM1* and *FLT3* mutation status (mutation-positive and mutation-negative) in GWAS 2 and GWAS 4. Data on *NPM1* and *FLT3* somatic mutation status was available for 653 and 865 AML cases, respectively, including 411 and 528 cases of cytogenetically normal AML, respectively. PCR mutation analysis was performed as part of routine diagnostics for *NPM1* exon 12 and *FLT3* exons 14-15, as previously described^{78, 79}.

Results

Given the identification of a major AML risk allele at the *HLA* locus on chromosome 6p21.32 and that cancer cells acquire somatic mutations that can function as neo-antigens for immune recognition we performed a case-control analysis stratified by mutation status for *NPM1* and *FLT3*, two genes commonly mutated and clinically significant in cytogenetically normal AML². Specifically, data on *NPM1* and *FLT3* somatic mutation status were available for 653 and 865 AML cases, respectively. There was no significant heterogeneity in AML risk when cases and controls were stratified by either *NPM1* or *FLT3* mutation status, although there was a trend towards higher risk for *NPM1*-mutated AML (OR 1.96, 95% CI 1.29 –2.98; $P = 1.7 \times 10^{-3}$) and *FLT3*-mutated AML (OR 1.52, 95% CI 1.07 –2.16; $P = 0.02$) compared to *NPM1*-wildtype AML (OR 1.28, 95% CI 0.97 –1.68; $P = 0.08$) or *FLT3*-wildtype AML (OR 1.26, 95% CI 1.01 –1.58; $P = 0.04$)(Supplementary Table 6).

Comment 3: In the last part of the results section, the authors state that they did not find any significant association with patient survival. Please comment, why data on outcome/treatment was only available in 767 out of 4413 AML patients?

Author response: Unfortunately, outcome data was not available on all of the AML cases recruited to this study. Rather, we have restricted the analysis to those cases where high quality data was available, including cases from UK clinical trials, the Study Alliance Leukemia in Germany and a hospital-based study in Hungary. We have applied for funding to expand the cohort of cases with high quality follow-up data that will increase statistical power allowing for analysis stratified by treatment and other factors.

Comment 4: Given the heterogeneous treatment of patients, I would have assumed that it will be difficult to identify loci associated with overall survival. However, to the reader it would be of great interest whether loci could be identified that are associated with response to cytarabine or daunorubicin, the standard chemotherapy backbone used in almost all patients who can be intensively treated. Previous reports in childhood AML suggested the presence of respective loci (Bargal et al. Oncotarget 2018).

Author response: This is a very good suggestion. Indeed, we have recently published a genome-wide association study for alleles that predict outcome in chronic lymphocytic leukaemia. We plan to apply the same methods to AML, but as yet we do not have sufficient cases with outcome data to justify a genome-wide analysis considering the stringent threshold required for publication ($P < 5 \times 10^{-8}$).

Minor comments:

Comment 5: In the abstract, the authors state that their findings “identify AML as a genetically heterogeneous malignancy”. In my opinion, this was already well known prior to this study. In accordance, I would suggest to rephrase this sentence to “confirm AML as a genetically heterogeneous malignancy”.

Author response: We agree with this reviewer. Due to space constraints we have moved this comment from the abstract to the discussion, as follows:

Our data **supports existing evidence of genetic and biological heterogeneity in AML²** and confirm the need for large collaborative studies to improve statistical power and aid the discovery of sub-type specific genetic risk loci.

We thank the reviewers for their excellent suggestions which have significantly strengthened our manuscript.

Reviewers' comments:

Reviewer #1 (Remarks to the Author):

The authors have address some of my concerns; however, I still have concerns about population stratification, which can lead to false positive findings. The authors used convenience controls from different populations and/or genotyped on different platforms. They have performed some additional quality control metrics and provided additional supplementary plots showing the principal components for GWAS1-GWAS4. However, the cases and controls still do not appear very well matched, as evident by the shift in cases relative to the controls seen on the PC plots. GWAS3 looks particularly poorly matched, and indeed, you can see evidence of systematic inflation for GWAS3 on the q-q plot with the red line above the black. For GWAS4, a subset of the cases form a distinct cluster separate from most controls. The authors stated that they excluded some subjects, but the PC plots still show a drift in cases relative to controls. The authors should either genetically match cases to controls, using PCAmatchR or similar program, or exclude the outlier clusters from each GWAS.

The suggestive findings should be removed from the abstract.

The numbers of cases/controls reported in text/abstract/figures do not always add up or agree.

Supplementary Figure 1. It would be helpful if the authors added the number of genotyped SNPs for each GWAS after QC metrics that were used for imputation

Supplementary Tables 3, 4 and 6 have lines designated as “tests for heterogeneity”, but no p-values are given for these tests. I would have expected the authors to test for heterogeneity between age <55 and >=55 or between males and females and put the p-value on this line. I am not sure why the lines are labeled but have no results.

Reviewer #2 (Remarks to the Author):

The authors have fully addressed all of my comments as well as the comments of the second reviewer.

Lin et al. Genome-wide association study identifies susceptibility loci for acute myeloid leukemia

Thank you for giving us the opportunity to respond to reviewers comments and revise our manuscript accordingly. We are pleased to see that reviewer 2 is happy with the revisions we have made and that reviewer 1 is happy with the majority of the revisions, although they still have concerns about the potential effects of population stratification. Reviewer 1 has also made a suggestion and identified a small number of technical errors. We have addressed the reviewers comments as detailed below and changes to the text are highlighted in red.

Reviewer 1, comment 1: The authors have address some of my concerns; however, I still have concerns about population stratification, which can lead to false positive findings. The authors used convenience controls from different populations and/or genotyped on different platforms. They have performed some additional quality control metrics and provided additional supplementary plots showing the principal components for GWAS1-GWAS4. However, the cases and controls still do not appear very well matched, as evident by the shift in cases relative to the controls seen on the PC plots. GWAS3 looks particularly poorly matched, and indeed, you can see evidence of systematic inflation for GWAS3 on the q-q plot with the red line above the black. For GWAS4, a subset of the cases form a distinct cluster separate from most controls. The authors stated that they excluded some subjects, but the PC plots still show a drift in cases relative to controls. The authors should either genetically match cases to controls, using PCAmatchR or similar program, or exclude the outlier clusters from each GWAS.

Author response:

We included nominally significant principal components as covariates in the analyses as a method to control the inflation, as previously suggested by reviewer 1. As such, we report low genomic inflation in all 8 analyses in our study (for all AML $\lambda_{GC}=1.021, 1.025, 1.055, 1.012$; and for cytogenetically normal AML $\lambda_{GC}=1.006, 1.011, 1.025, 1.001$), consistent with other GWAS published in Nature Communications. For the current iteration of our manuscript we had already removed outlying cases and controls to improve the case-control matching. However, we do not feel we could justify the removal of any additional cases because these occupy space on the PCA plots that also includes controls, and this is unlikely to further reduce the genomic inflation.

However, we recognise the reviewer's concern that population sub-structure across Europe might have impacted the associations reported in our study. In order to test this directly we excluded non-UK cases from GWAS1, GWAS2 and GWAS4 such that all cases and controls are of UK origin and have CEU ancestry for these three studies. GWAS3 is comprised almost exclusively of AML cases and controls from continental Europe. PCA plots for GWAS1, GWAS2 and GWAS4 restricted to UK cases and controls with CEU ancestry profiles are shown below:

PCA plots showing PC1 and PC2 for GWAS1, GWAS2 and GWAS4 restricted to UK AML cases (red circles) and UK controls (blue triangles) with CEU ancestry. Case:control numbers are 869:2671 for GWAS1, 512:2477 for GWAS2 and 775:3728 for GWAS4.

We subsequently re-analysed the data for GWAS1, GWAS2 and GWAS4 restricted to UK cases and controls with CEU ancestry. We have also included the data for GWAS3 in the meta-analysis in order to facilitate a direct comparison with the “all European” analysis presented in our manuscript, as follows:

Chromosome 11 (*KMT5B*:rs4930561)

All European AML cases

UK-only cases and controls

Chromosome 6 (*HLA*:rs3916765)(cytogenetically normal AML)

All European AML cases

UK-only cases and controls

Chromosome 1 (CACHD1:rs10789158)(cytogenetically normal AML)

All European AML cases

UK-only cases and controls

Chromosome 7 (AKR1B1:rs17773014)

All European AML cases

UK-only cases and controls

The direction and magnitude of the association results are very similar between the UK-only analysis and the original European analysis. As expected, there is a reduction in statistical significance for some of the signals in the UK-only analysis due to the reduced statistical power, although there is a slight increase in significance for the chromosome 1 signal.

We have not included results from the UK-only analysis in our revised manuscript but would be happy to add these should the reviewer or editors feel that this would strengthen our manuscript.

Reviewer 1, comment 2: The suggestive findings should be removed from the abstract.

Author response: We have removed the borderline significant associations from the manuscript, which now reads as follows:

Acute myeloid leukemia (AML) is a hematological malignancy with an undefined heritable risk. Here we performed a meta-analysis of three genome-wide association studies, with replication in a fourth study, incorporating a total of 4018 AML cases and 10488 controls. We identified a genome-wide significant risk locus for AML at 11q13.2 (rs4930561; $P = 2.15 \times 10^{-8}$; *KMT5B*). We also identified a genome-wide significant risk locus for the cytogenetically normal AML sub-group (N=1287) at 6p21.32 (rs3916765; $P = 1.51 \times 10^{-10}$; *HLA*). Our results inform on AML etiology by identifying putative functional genes operating in histone methylation (*KMT5B*) and immune function (*HLA*).

We have also modified the title to remove the word “four”, which now reads as follows:

Genome-wide association study identifies susceptibility loci for acute myeloid leukemia

We have also modified the last sentence of the introduction to reflect this change, which now reads as follows:

Here we report the identification of risk loci AML, including pan-AML irrespective of disease sub-type and for cytogenetically normal AML. These data inform on disease etiology and demonstrate the existence of common, low-penetrance susceptibility alleles for AML with heterogeneity in risk across sub-types.

Reviewer 1, comment 3: The numbers of cases/controls reported in text/abstract/figures do not always add up or agree.

Author response: We apologise for these errors which we have now corrected in the revised manuscript. Specifically, the total number of AML cases is now correctly shown as 4018 in the abstract and main text of the article (see above). Likewise, the total number of AML cases and controls from the discovery analysis is now correctly shown as 3014 and 6760, respectively, in the main text of the article. Finally, the correct number of cytogenetically normal AML cases with survival data available is now correctly listed as 369 in the main text of the article.

Reviewer 1, comment 4: Supplementary Figure 1. It would be helpful if the authors added the number of genotyped SNPs for each GWAS after QC metrics that were used for imputation

Author response: We have modified Supplementary Figure 1 to include the number of genotyped SNPs for each GWAS that were used for the imputation. We thank the reviewer for this suggestion.

Reviewer 1, comment 5: Supplementary Tables 3, 4 and 6 have lines designated as “tests for heterogeneity”, but no p-values are given for these tests. I would have expected the authors to test for heterogeneity between age <55 and >=55 or between males and females and put the p-value on this line. I am not sure why the lines are labeled but have no results.

Author response: We erroneously uploaded an incomplete version of the Supplementary Tables file to the Nature Communications portal. Please accept our apologies for the confusion this caused. We have now uploaded the correct version that includes the r^2 values and the test for heterogeneity P values (PQ).

We thank the reviewers for their excellent suggestions which have significantly strengthened our manuscript.

Reviewers' comments:

Reviewer #1 (Remarks to the Author):

The authors have addressed many of my comments, but neglected to address the poorly matched controls and systemic inflation seen in GWAS 3. Just because the cases and controls are all from continental Europe does not mean that they are free of population substructure. The authors provided a sensitivity analysis where they limited the analysis to cases and controls from the UK for GWAS 1, 2 and 4. I am not sure why they did not limit GWAS 3 to cases and controls from German as part of that same sensitivity analysis. It would be helpful for the reader if they provided the results of the sensitivity analysis (limited to UK cases and controls for GWAS 1,2, and 4 and limited to German cases and controls for GWAS 3) as a supplementary figure.

Lin et al. Genome-wide association study identifies susceptibility loci for acute myeloid leukemia

Thank you for giving us the opportunity to respond to reviewers comments and revise our manuscript accordingly. We have addressed the reviewers comment as detailed below:

Reviewer 1, comment 1: The authors have addressed many of my comments, but neglected to address the poorly matched controls and systemic inflation seen in GWAS 3. Just because the cases and controls are all from continental Europe does not mean that they are free of population substructure. The authors provided a sensitivity analysis where they limited the analysis to cases and controls from the UK for GWAS 1, 2 and 4. I am not sure why they did not limit GWAS 3 to cases and controls from German as part of that same sensitivity analysis. It would be helpful for the reader if they provided the results of the sensitivity analysis (limited to UK cases and controls for GWAS 1,2, and 4 and limited to German cases and controls for GWAS 3) as a supplementary figure.

Author response: We thank the reviewer for this excellent suggestion and we have now re-analysed the data for GWAS 3 restricted to German cases and controls of CEU origin. PCA plots for the German cases/controls in GWAS 3 and UK cases in GWAS 1, GWAS 2 and GWAS 4 are included in the revised manuscript (Supplementary Figure 12), as follows:

Supplementary Figure 12: PCA showing PC1 and PC2 restricted to CEU cases and controls of UK origin for (A) GWAS 1, (B) GWAS 2 and (D) GWAS 4 and restricted to CEU cases of German origin for (C) GWAS 3.

We have also included the German data from GWAS 3 in a meta-analysis with the UK data from GWAS 1, GWAS 2 and GWAS 4 and the results are included in the revised manuscript (Supplementary Figure 13), as follows:

Supplementary Figure 13: Forest plots for 4 new loci associated with acute myeloid leukemia restricted to CEU cases of UK (GWAS 1, 2 and 4) or German (GWAS 3) origin. Study cohorts, sample sizes (case and controls (con)), imputation (info) score, effect allele, effect allele frequencies (EAF) and estimated odds ratios (OR) for rs4930561 (A), rs3916765 (B), rs10789158 (C) and rs17773014 (D). The vertical line corresponds to the null hypothesis (OR=1). The horizontal lines and square brackets indicate 95% confidence intervals (95% CI). Areas of the boxes are proportional to the weight of the study. Diamonds represent combined estimates for fixed-effect and random-effect analysis. Cochran's Q statistic was used to test for heterogeneity such that $P_{HET} > 0.05$ indicates the presence of non-significant heterogeneity. The heterogeneity index, I^2 (0-100) was also measured which quantifies the proportion of the total variation due to heterogeneity.

The direction and magnitude of the association results are very similar between the UK/Germany-only meta-analysis and the original European meta-analysis (Figure 2). As expected, there is a reduction in statistical significance for some of the signals in the UK/Germany-only analysis due to the reduced statistical power, although there is a slight increase in significance for the chromosome 1 signal.

We have referred to the new supplementary figures in the results section of the revised manuscript, as follows:

Meta-analysis of SNPs common to all four GWAS (N=6661818 and N=6496414 for all AML and cytogenetically normal AML, respectively) also revealed additional borderline significant susceptibility loci at 1p31.1 (rs10789158, *CACHD1*, $P = 2.25 \times 10^{-7}$) for all AML and at 7q33 (rs17773014, *AKR1B1*, $P = 4.09 \times 10^{-7}$) for cytogenetically normal AML, both with consistent

direction and magnitude of effect across all four studies (**Figures 1 and 2**). In order to test for any potential effects of residual population substructure we re-examined the top signals using only cases and controls of UK origin in GWAS1, GWAS2 and GWAS4, and using only cases and controls of German origin in GWAS3 (Supplementary Figure 12). The direction and magnitude of the associations in these sub-group analyses (Supplementary Figure 13) are very similar to the analyses including all cases and controls (Figure 2). As such, we report genome-wide significant susceptibility loci for all AML and cytogenetically normal AML at 11q13.2 (rs4930561) and 6p21.32 (rs3916765), respectively, and borderline significant susceptibility loci for all AML and cytogenetically normal AML at 1p31.1 (rs10789158) and 7q33 (rs17773014).

We thank the reviewers for their excellent suggestions which have significantly strengthened our manuscript.

Reviewers' comments:

Reviewer #1 (Remarks to the Author):

The authors have addressed my concerns.